# Cohesive Zone Modeling of Pull-Out Test for Dental Fiber–Silicone Polymer

**DOI:** 10.3390/polym15183668

**Published:** 2023-09-06

**Authors:** Ayman M. Maqableh, Muhanad M. Hatamleh

**Affiliations:** 1School of Electro-Mechanical Engineering, Luminus Technical University College (LTUC), Amman 11118, Jordan; 2Faculty of Applied Medical Science, Allied Dental Sciences Department, Jordan University of Science and Technology (JUST), Irbid 22110, Jordan; muhanad.hatamleh@gmail.com

**Keywords:** FEA, FRC, fiber-reinforced composite

## Abstract

Background: Several analytical methods for the fiber pull-out test have been developed to evaluate the bond strength of fiber–matrix systems. We aimed to investigate the debonding mechanism of a fiber–silicone pull-out specimen and validate the experimental data using 3D-FEM and a cohesive element approach. Methods: A 3D model of a fiber–silicone pull-out testing specimen was established by pre-processing CT images of the typical specimen. The materials on the scans were posted in three different cross-sectional views using ScanIP and imported to ScanFE in which 3D generation was implemented for all of the image slices. This file was exported in FEA format and was imported in the FEA software (PATRAN/ABAQUS, version r2) for generating solid mesh, boundary conditions, and material properties attribution, as well as load case creation and data processing. Results: The FEM cohesive zone pull-out force versus displacement curve showed an initial linear response. The Von Mises stress concentration was distributed along the fiber–silicone interface. The damage in the principal stresses’ directions S11, S22, and S33, which represented the maximum possible magnitude of tensile and compressive stress at the fiber–silicone interface, showed that the stress is higher in the direction S33 (stress acting in the Z-direction) in which the lower damage criterion was higher as well when compared to S11 (stress acting in the XY plane) and S23 (stress acting in the YZ plane). Conclusions: The comparison between the experimental values and the results from the finite element simulations show that the proposed cohesive zone model accurately reproduces the experimental results. These results are considered almost identical to the experimental observations about the interface. The cohesive element approach is a potential function that takes into account the shear effects with many advantages related to its ability to predict the initiation and progress of the fiber–silicone debonding during pull-out tests. A disadvantage of this approach is the computational effort required for the simulation and analysis process. A good understanding of the parameters related to the cohesive laws is responsible for a successful simulation.

## 1. Introduction

Restoring significant facial defects often necessitates the use of a facial prosthesis, which can be constructed from silicone, acrylic resin, or a combination of both materials [1,2]. Various issues have been reported by different researchers that can affect the functionality and effectiveness of these facial prostheses. These problems encompass issues like silicone degradation, separation of silicone from the acrylic base, limitations in simulating natural facial expressions due to the stiffness of the acrylic base, and a reduction in the seamless integration of the prosthesis, resulting in visible gaps [3,4,5,6]. 

Efforts have been made to develop new polymeric materials that possess improved mechanical properties, including a higher tear strength, lower hardness, and reduced viscosity [6,7,8]. It has also been proposed that facial prostheses be fabricated in three layers, comprising a silicone rubber base material, an inner silicone gel layer, and a thin outer polymeric coating. However, attempts to modify materials to address specific shortcomings have often resulted in the compromise of other desirable characteristics. While existing materials for facial prostheses have seen enhancements, they are not yet considered ideal [9]. Ongoing research aims to create prostheses composed of two or more materials, laminated and bonded together, each possessing its own optimal properties [9].

Recently, there has been a growing interest in utilizing fiber-reinforced composite (FRC) as a biomaterial in dental and medical applications [10,11,12,13]. When applied to maxillofacial silicone prostheses, FRC tends to provide more stable margins and a strong bond between the silicone and the fibers. Moreover, the silicone component offers a realistic sensation when in contact with the skin, meeting the aesthetic requirements of patients and enhancing their quality of life [11]. The new technique reported indicated that the silicone polymer can be encapsulated in a retentive glass fiber-embedded framework as in fiber-embedded maxillofacial prostheses [11]. 

Fiber-reinforced composites, particularly those incorporating continuous glass fibers, offer a wide range of mechanical properties, dependent on factors such as the fiber type, orientation, quantity, and polymer matrix [14,15]. They have been proved to be suitable dental and medical biomaterials [16,17]. 

During clinical service, forces expected to influence the bond integrity between fibers and silicone polymers are likely to be generated when the patient holds the silicone to dislodge the prosthesis from the magnetic retentive sites or bars, or during cleaning of the prosthesis [18]. 

The pull out forces required to disrupt the bond integrity between fibers and silicone elastomer have been reported previously to be in the range of (12–20 N). However, earlier reports have indicated various difficulties with FRC bonding including the experimental handling of samples and/or the ability to observe the fracture events instrumentally or usually with enough detail [19,20,21,22]. 

Several analytical methods, such as the fragmentation test, microbond test, and fiber pull-out test have been developed to evaluate bond strength of fiber–matrix systems [20,23]. The single fiber pull-out test has been extensively used due to its importance in understanding the fiber–matrix interface, stress distribution, and strength within materials [15]. The bond strength is determined by measuring the force required to pull-out a fiber bundle embedded in the matrix [19,20,24,25,26]. Photoelasticity has been also used to experimentally measure the stress field at the fiber–matrix interface [27].

One of the most recent numerical methods to investigate fracture mechanics is the finite element method (FEM). This method is based on continuum damage theories, in which the difficulty in relating its parameters to well-defined experimental parameters (i.e., fracture energy) is pointed out. In addition, these methods do not consider the discrete nature of fracture mechanics [27,28]. A new FEM alternative for these theories has been related as a discrete approach based on the cohesive elements used at the interface between standard volume elements to nucleate cracks and propagate them following the deformation process [29,30,31,32,33,34]. The aim of the study was to investigate the debonding mechanism of a fiber–silicone pull-out specimen and validate the experimental data achieved using 3D-FEM and a cohesive element approach.

## 2. Material and Methods

### 2.1. Specimen Construction and Test Methodology

The construction process was previously described [15]. Three test samples were created by embedding unidirectional glass fiber bundles (manufactured by C&B Fibers, StickTech, Turku, Finland) with a diameter of 1.5 mm and an embedded length of 20 mm into a heat-polymerized silicone elastomer (Cosmesil M511, Principality Medical, Newport, UK). The construction of these specimens involved the use of a two-part sectional flask measuring 100 mm × 80 mm × 30 mm. The lower section of the flask had three holes with a diameter of 1.50 mm and a depth of 5 mm, into which the fiber bundles were securely fixed. Meanwhile, the upper section contained three cylindrical-shaped molds measuring 14.40 mm in diameter and 20 mm in length, where the silicone material was packed. The two parts of the flask were separated by a thin layer of sodium alginate (Hillier Dental, Kent, UK).

The glass fiber bundles were subjected to light polymerization for 4 min using a curing unit (ESPE visio ^®^ Beta vario, 3M ESPE, Seefeld, Germany). Subsequently, the second part of the flask was assembled over the basal part, allowing the fiber bundles to protrude through the center of the cylindrical molds. The maxillofacial silicone elastomer was prepared according to the manufacturer’s instructions, using a ratio of 10 g of rubber to 1 g of hardener, which was measured with a micro-balance. The silicone was manually mixed for 5 min and then mechanically mixed under vacuum conditions for 5 min using a Multi Vac 4 (Degussa, Munich, Germany).

After the mixing process was completed, the silicone was poured into the flask molds with the assistance of vibration. The flask contents were heat-polymerized in an oven (Gallenkamp, London, UK) in accordance with the manufacturer’s guidelines, which involved heating at 100 °C for 1 h. Following this, the specimens were gently removed and allowed to cool at room temperature for 2 h. Subsequently, the specimens were carefully taken out and stored in a dry environment for 24 h.

A universal testing machine was utilized to undergo the mechanical test of pulling the fibers out the silicone matrices according to the international society organization standard (ISO 3501 Tensile Pull Out Test) [35]. A flat rectangular-shaped, metal plate (1.50 mm thick, 16 mm wide, and 20 mm long) was placed on the top of each specimen [11]. The plate had a middle opening (5.30 mm in diameter) through which the free parts of the fibers (which were 5 mm long) were pointing out. Then, the free part was fixed in the chuck of the upper member of the testing machine. The periphery of the metal plate was clasped between the grips of a holder which is fixed in the base of the testing machine. Since the metal plate was wider than the diameter of the specimen (16 mm and 14.20 mm), it clasped the specimen in place and prepared it for running the test without exerting any radial stresses on the silicone matrices (Figure 1 and Figure 2). The specimens (n = 3) were tested and all specimens were fixed in alignment parallel to the long axis of the testing machine so that no bending movements were created in performing the test. The specimens were tested at room temperature (23 °C ± 1 °C).

### 2.2. Finite Element Modeling: The Steps of FEM Modeling Are Shown in Figure 3 and Figure 4

Pre-processing of CT image slices: The pre-processing of this study was performed to build a 3D model of a typical fiber–silicone pull-out testing specimen as shown in Figure 4. Firstly, 866 CT image slices (19.5 µm thick/each slice) were obtained by using a high resolution Micro-CT (µCT, SkyScan 1072, Aarstselaar, Belgium). Secondly, the materials on the scans were posted in three different cross-sectional views using ScanIP (Simpleware, Version U-2022.12) in which masks were applied for the different materials to allow extended visualization based on image density thresholding.

As a result of the segmentation of these masks, the ScanIP file was imported to ScanFE in which a 3D generation was implemented for all of the image slices. This file was exported in FEA format to be processed in ABAQUS. The file was imported in the finite element analysis and was re-meshed. The mesh was improved using Hypermesh 8.0 (Altair Hyperworks), because the congruence was lost during the previous remeshing process. The optimized STL files were imported in a finite element analysis software package (PATRAN/ABAQUS, version r2) for the generation of the solid mesh, boundary conditions, and material properties attribution, as well as load case creation and data processing. 

**Figure 3 polymers-15-03668-f003:**
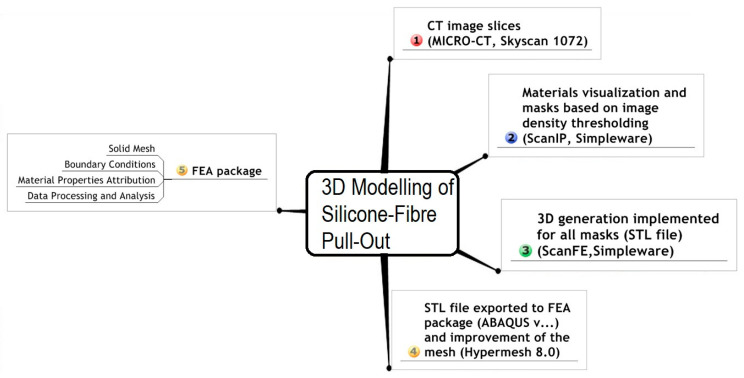
Steps of the 3D pull-out specimen modeling.

**Figure 4 polymers-15-03668-f004:**
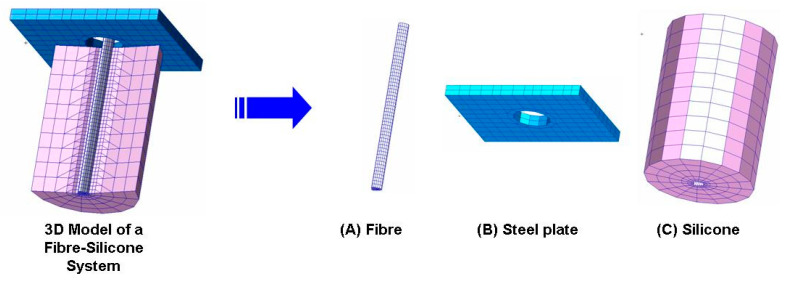
3D model of a typical fiber–silicone pull-out testing specimen.

The solid mesh was performed by eight-node hexahedral elements and the debonding propagation was performed by using cohesive elements which are user-defined elements within ABAQUS. The total energy for a traction-separation damage evolution (G_c_) was 0.11 × 10^−3^ J and the stress criteria for the cohesive approach S_11_, S_22_, and S_33_ were 5, 5, and 0.15 MPa, respectively (Table 1). 

### 2.3. Boundary Conditions, Material Properties and Data Processing

The boundary conditions applied to the model are represented in Figure 5. From the experimental pull-out test, the fiber displacement of load applied and the corresponding pull-out force are known for each load step.

No friction was assumed for the debonded region in the present study. Excluding the cohesive interface, which is anisotropic, it was assumed that all of the materials were homogeneous, isotropic, linear, and elastic and that all the interfaces were perfectly bonded and could transfer shear and normal stresses. The cohesive interface needs to be anisotropic because the cohesive elements need to be softened in the directions S11 and S22; decreasing the reaction force of them in these directions will enable the fiber to be pulled out from the silicone matrix. The properties of the materials are presented in Table 2. 

The Von Mises stress distribution of the fiber–silicone specimen was investigated and the maximum shear stress distribution was plotted to investigate the stress concentration during the debonding process at the cohesive interface (for the stress criteria of each direction: S11, S22, and S33).

## 3. Results 

A typical experimental pull-out curve of 20 mm embedded length is shown in Figure 6. Force is plotted against displacement (linear region), exhibiting a single peak where the debonding took place, and then followed by lower peaks indicating frictional forces between the fiber and the matrix. The test terminated when the fiber was fully pulled out from the silicone matrix. In the initial quasi-linear region (A), the fiber extends as the force rises. At Point I, the shear damage initiates. When the maximum force D (debonding force) is reached, the fiber debonds from the matrix along the full embedded length. At Point II, a region of traction (T) initiates as a sudden drop of the force E (initial extraction force) required for pulling out the debonded fiber from the matrix. Finally, the force continues to decrease as successive lower peaks at the friction region (F), while the extracted length increases until the fiber is totally extracted from the matrix (Point III). The same curve is shown in Figure 7 with more details of the debonding region. For this curve, the displacement was plotted until a 1.8 mm implantation length of the fiber in the silicone matrix. 

A FEM cohesive zone pull-out force versus displacement is shown in in Figure 8. The displacement was plotted until the debonding failure. The curve shows an initial linear response where the fiber extends as the force rises (quasi-linear region A) up to a peak load D (or in this case, to the apparent strength) when the fiber debonds from the silicone, followed by an exponential strain-softening region, after a drop of the curve to the initial extraction (I) (required for pulling out the debonded fiber from the silicone). The force continues to decrease at region T until the whole fiber is extracted from the silicone at 1.5 mm of displacement (Point III). After this region (T), the tensile stresses are higher than the shear and the displacement jumps across the interface. From the intrinsic cohesive failure model, as the interface separates, the magnitude tensile stress in the T region (the apparent strength) reaches a maximum, after which it progressively becomes null, meaning that the fiber–silicone is completely debonded. 

The Von Mises stress concentration zone at the interface near the fiber is shown in Figure 9. At the region with the red fringe, the maximum shear stress concentration is represented. The stress is distributed along the interface and also at the end of the matrix. The damage in the three directions (S11, S22, and S33) for the cohesive zone (interface) is shown in Figure 10, Figure 11 and Figure 12, respectively. The stress is higher in the direction S33 in which the lower damage criterion was higher as well (0.15 MPa) in comparison to the others (both 5 MPa). The initial debonding of the fiber from the silicone matrix is shown on Figure 13. The cohesive elements are located between the materials, representing the interface cohesive zone.

## 4. Discussion

The single-fiber pull-out test has been pointed out by many authors as a simple method that can be used to evaluate interfacial properties such as interfacial shear strength and frictional stress [20,24]. Most of them have used the interfacial shear strength as a criterion for fracture, i.e., a crack will propagate when the interfacial shear stress at a crack tip is greater than the shear strength of the interface. On the other hand, the debonding process can also be treated using the fracture energy as a failure criterion. According to Griffith’s fracture criterion, the debonding will take place when the work done by the applied load minus the energy stored in the system is larger than the work of the interfacial detachment, which is the adhesive fracture energy. This energy is the amount of energy to separate the unit area of the interface and is used to characterize the bonding strength. 

Some studies have pointed out that during the fiber pull-out tests, initial debonding, partial debonding, and complete debonding at the interface occur sequentially and the initial debonding and complete debonding are respectively the beginning and the end of the partial debonding [36,37]. They have reported that after initial debonding, the displacement increases with the increasing stress initially and before the displacement reaches its maximum value, the stress exhibits a decrease and the debonding occurs when the displacement reaches the maximum value. The experimental curves obtained in the present study are in agreement to these theories. In addition, they could indicate that the free part of the fiber outside the matrix is loaded to a point where partial debonding happened (I) and that part is free from the stress, causing an instantaneous drop of force to point E (initial extraction force) [38].

Recent studies have used numerical analysis to understand the stress distribution at the pull-out set-up. The photoelasticity technique has been used to investigate the stress field in a matrix at a fiber interface by calculating the shear stress profile and pointing out the location of micromechanical events [27], but this method cannot reach the complexity of the interface stress distribution in detail. The finite element method has also been used to investigate the single pull-out test; however, most of these studies were based on few parameters or on friction laws that still have known difficulties in understanding and simulating the interface behavior [23,24,25,28,39].

Most recently, the FEM was used to carry out the pull-out test simulation using the code ABAQUS^®^ and its cohesive elements approach, which is characterized by interface elements with a cohesive constitutive model [30,31,32,40]. The interface elements were inserted between the fiber and the silicone matrix for representation of the bonded area and the debonding process. This cohesive approach includes the complex behavior of debonding which is a function of the fracture energy (critical energy release rate G_c_ or ‘cohesive energy’) and the strength of the interface (peak strength σ_c_ or ‘cohesive strength’). For this type of simulation, an initial crack is not essential [32]. The separation of the cohesive interfaces is calculated from the difference of the displacements of the adjacent continuum elements [32]. There is a linear relationship that terminates when initial debonding occurs. After initial debonding, the stress increases at a slower rate with the increase in the displacement, and it reaches a maximum (in this study, this point is determined by the maximum shear stress, σ_c_) when complete debonding occurs along the full embedded fiber length (one of the components of the displacement vector reaches a critical value (δ_c_)) and the continuum elements initially connected by this cohesive element are disconnected, which means that the failure has occurred. 

In Figure 6, Point I represents the failure in which the cohesive elements have lost their stiffness so that the continuum elements are disconnected. Since the solid elements overestimate the maximum shear stress, the failure of the cohesive elements is promoted and this was well represented by the simulation at around 0.75 mm of displacement, which confirms the potential of this method to assess the predictive sensitivities of structural damage tolerance. This prediction can be confirmed in Figure 5, which shows the point D with a higher value (1.60 mm) than the corresponding value in Figure 6. The maximum shear stress is achieved when the value of the normal stress (11, 22, and 33 direction), first shear stress, or second shear stress reaches the limit determined by the stress criteria (S_1_: 5 MPa; S_2_: 5 MPa, and S_3_: 0.15 MPa). At this point, the stress drops to a new value and the defect responsible for fiber pull-out initiates. Then, the stress continuously decreases to zero until the fiber is fully pulled out of the silicone.

It is already known that the friction took place after the debonding between the fiber and matrix had happened, which supports the absence of this parameter in this study which is focused on the debonding that happens when the maximum shear stress is achieved. The differences between the two curves after the maximum shear stress are attributed to the fact that the friction was not simulated in this study. After the single peak at point D in Figure 4, where debonding took place, there were lower successive peaks indicating the presence of frictional forces between the fiber and the matrix. Since the cohesive interface model tends to “smear” the localized effects near the crack tip through gradual development of the displacement jumps and gradual decay of the tractions, these peaks are not described by the cohesive interface model [41]. In Figure 6, this region is represented by a quasi-linear region with some step-wise (T) where traction is acting almost without any obstacle and this is why the decrease is progressive until the full pull-out of the fiber from the silicone matrix instead of presenting successive peaks. 

The step-wise characteristic at this region might happen as a function of the anisotropy of the cohesive interface which promotes a strain-softening region in the curve (Figure 6). The results from the shear stress distribution in the softened directions (1 and 2) of the present study in Figure 8 and Figure 9 are in accordance with some authors that have shown that the damage in one direction is not independent of what happens in the other [40], which means that the fiber–silicone system represented could support shear and tensile stress in a combined way. In this type of simulation, the cohesive interface needs to be anisotropic because the cohesive elements need to be softened in Directions 1 and 2 to enable the fiber to be pulled out from the silicone matrix. The shear stress distribution in Direction 3 with the high elastic modulus (10 MPa) and the lowest damage criterion (0.15 MPa) is shown in Figure 10. And this represents the highest stress concentration, as Figure 11 shows that when the effective traction acting on the elements’ facets reaches the cohesive strength of the material (maximum shear stress, σ_c_), the cohesive elements are inserted adaptatively at their interior and this means that they are disconnected.

## 5. Conclusions

The comparison between the experimental values and the results from the finite element simulations show that the proposed cohesive zone model accurately reproduces the experimental results. These results are considered almost identical to the experimental observations about the interface. The FEA simulation is in good agreement with the experimental results in the part before the peak load, but the agreement is not as promising in the softening part. The cohesive element approach is a potential function that takes into account the shear effects with many advantages related to its ability to predict the initiation and progress of the fiber–silicone debonding during pull-out tests. Among the disadvantages of this approach are the computational efforts required for the simulation and analysis process. A good understanding of the parameters that are related to the cohesive laws is responsible for a successful simulation.

## Figures and Tables

**Figure 1 polymers-15-03668-f001:**
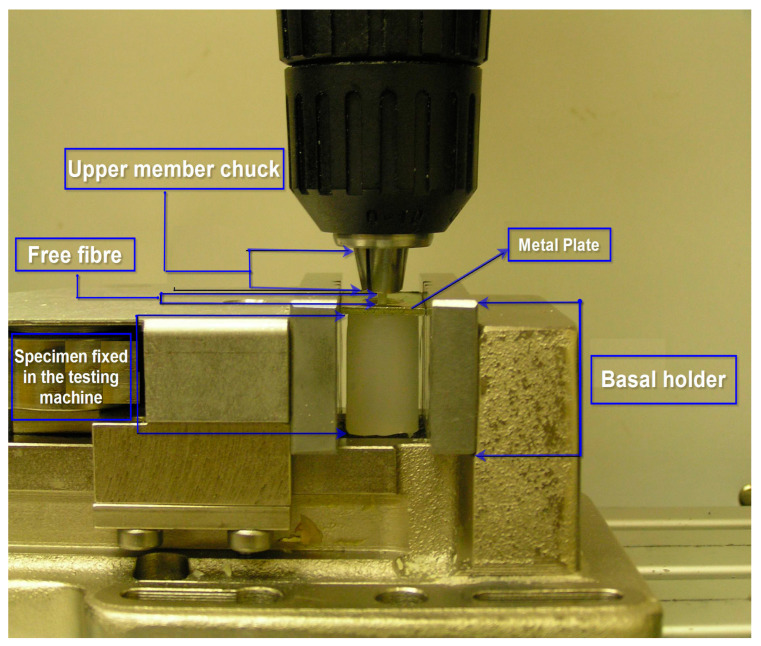
A graph showing how specimens were fixed between the grips of the testing machine.

**Figure 2 polymers-15-03668-f002:**
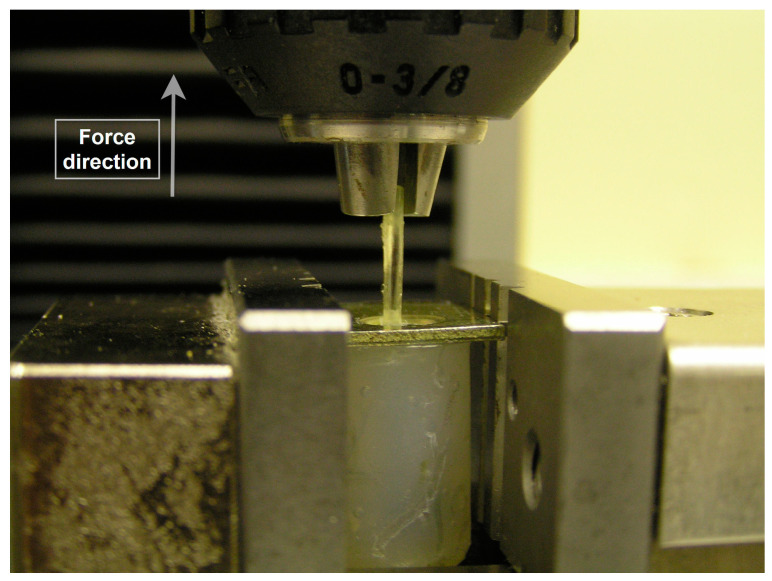
A graph of an ongoing pulling of a fiber out a silicone body at a 1 mm/min pulling speed.

**Figure 5 polymers-15-03668-f005:**
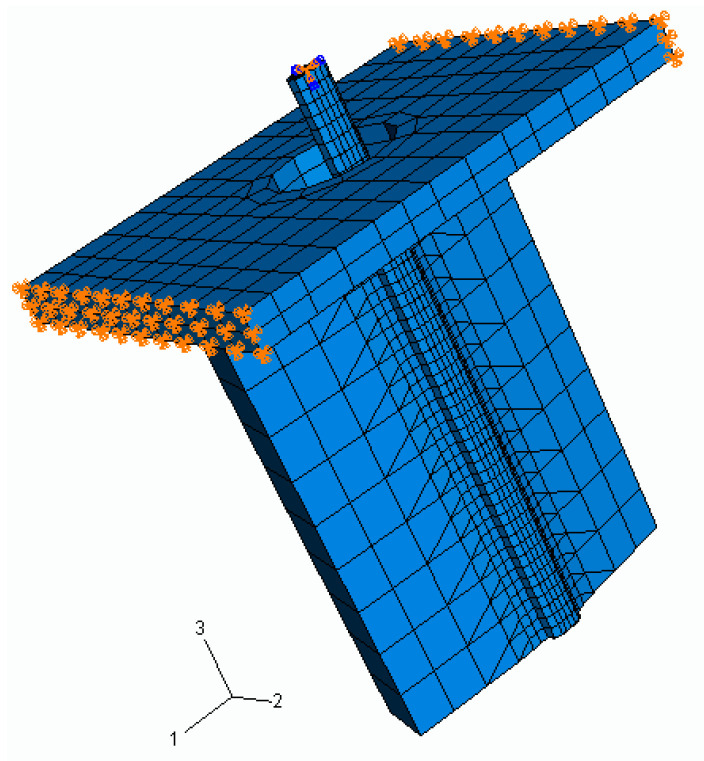
The boundary conditions applied to the model (appearing in orange colour).

**Figure 6 polymers-15-03668-f006:**
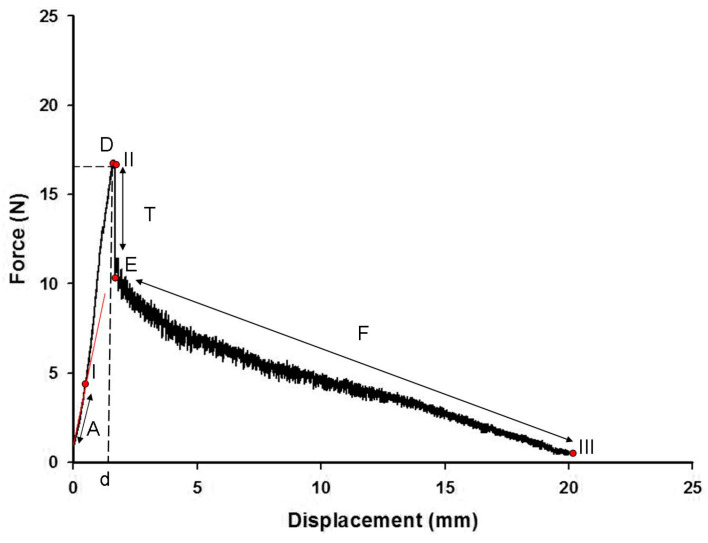
Experimental pull-out force-displacement curve of 20.0 mm embedded length. (A) is the quasi-linear region; (I) is the initial shear damage point; (D) is the maximum force; (d) is the displacement of debonding; (II) is the initial traction point; (T) is the region of traction; (E) is the initial extraction force; (F) is the region of friction; and (III) is the point of complete debonding.

**Figure 7 polymers-15-03668-f007:**
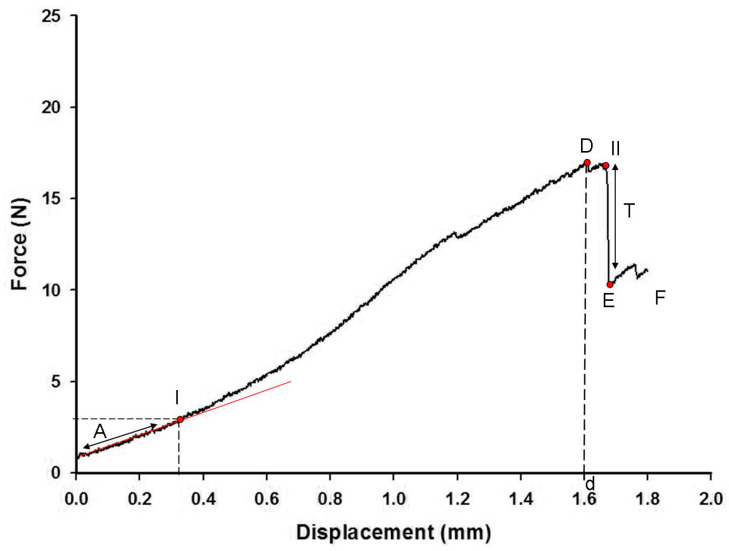
Experimental pull-out force-displacement curve for 1.8 mm implantation length. (A) is the quasi-linear region; (I) is the initial shear damage point; (D) is the maximum force; (d) is the displacement of debonding; (II) is the initial traction point; (T) is the region of traction; (E) is the initial extraction force; and (F) is the region of friction.

**Figure 8 polymers-15-03668-f008:**
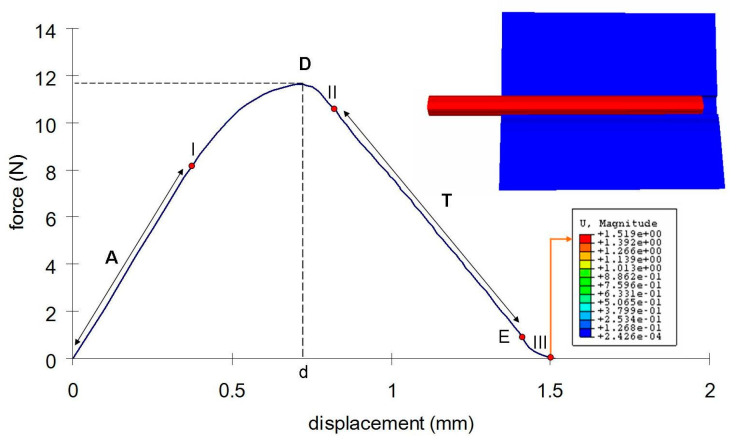
Apparent strength vs. displacement response of a fiber–silicone specimen simulated by the finite element method. (A) is the quasi-linear region; (I) is the initial shear damage point; (D) is the maximum force; (d) is the displacement of debonding; (II) is the initial traction point; (T) is the region of traction; (E) is the initial extraction force and (III) is the point of complete debonding.

**Figure 9 polymers-15-03668-f009:**
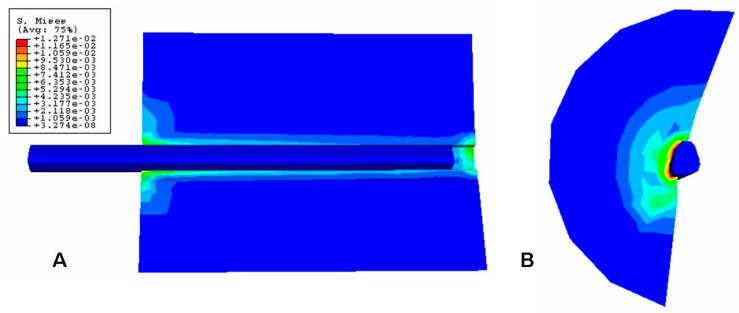
Von Mises stress of the fiber–silicone pull-out specimen simulation. (**A**) is a cross-sectional representation of the model (plane 1 0 3) and (**B**) is the top view of the model (plane 1 2 0).

**Figure 10 polymers-15-03668-f010:**
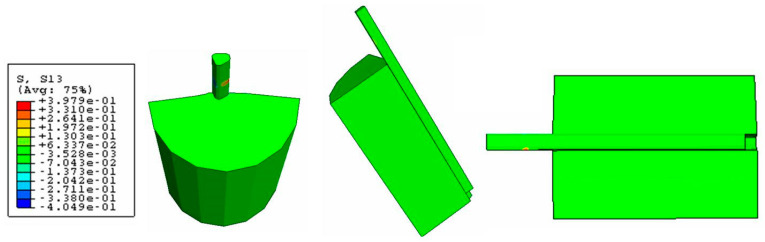
Damage stress distribution at the cohesive zone (S11 direction).

**Figure 11 polymers-15-03668-f011:**
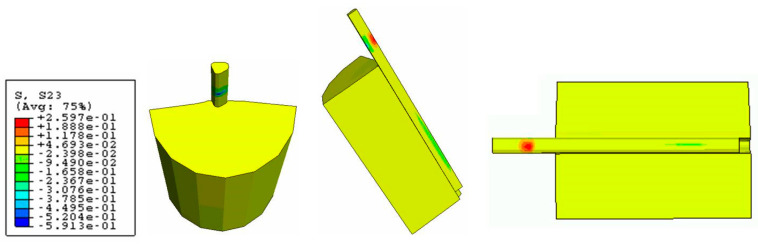
Damage stress distribution at the cohesive zone (S22 direction).

**Figure 12 polymers-15-03668-f012:**
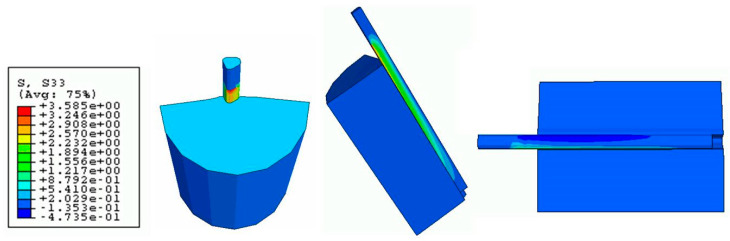
Damage stress distribution at the cohesive zone (S33 direction).

**Figure 13 polymers-15-03668-f013:**
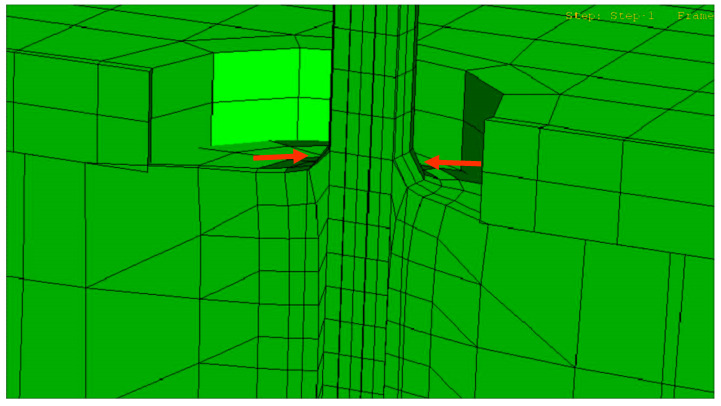
Cross-sectional virtual view showing the fiber–silicone interface. Initial debonding of the fiber from the silicone matrix (arrowed). The cohesive elements are located between the materials, representing the interface cohesive zone.

**Table 1 polymers-15-03668-t001:** Stress analysis in the Cartesian coordinate system.

Stress	Direction
S11	Normal stress component acting along the *X*-axis
S22	Normal stress component acting along the *Y*-axis
S33	Normal stress component acting along the *Z*-axis

**Table 2 polymers-15-03668-t002:** Materials properties.

Material	Elastic Modulus (MPa)	Poisson’s Ratios
Silicone	1.90	0.40
Fiber	81.300	0.2
Steel	210.000	0.30
Cohesive interface	0.50 (E_1_) 0.50 (E_2_) 10.0 (E_3_)	0.5

## Data Availability

No new data were created.

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
