# Peer review of "Cohesive Zone Modeling of Pull-Out Test for Dental Fiber–Silicone Polymer"

_polymers, 2023, doi:10.3390/polym15183668_

Round 1

Reviewer 1 Report

silicone polymer can encapsulate in a retentive glass fiber-embedded framework as in fiber-embedded maxillofacial prostheses. The authors had a study in this area. They tested the debonding mechanism of a fibre-silicone pull-out specimen and validate the experimental data achieved using 3D - FEM and a cohesive element approach. The study is novel and has main importance in its area. The introduction provides sufficient background and includes all relevant references. All the cited references are relevant to the research.

However, I have some comments;

The conclusion section of the abstract does not support the results. Please improve it.

the introduction section is so short. 

The sentence in lines 60-61 needs a reference.

Please mention the sample size determination.

Please mention the ISO number or standards for Sample Preparation.

The quality of Figure 3 s is low. please improve.

What were the statistical approaches?

More explanation are needed for figure 13.

The discussion section is so long and confusing. Please improve. 

The conclusion section is so brief and does not support the results. Please improve.

Minor editing of English language required.

Reviewer 2 Report

In the summary, you have to define the addresses S13, S23 and S33.

The degrees in temperature are not well indicated (do not use º, it is not the correct symbol)

In header 2.2. it is not the right place to refer to Figure 3.

In Figure 3, the underscores that are not part of the image should be removed.

On page 3, line 97, reference is made to stresses S1, S2 and S 3. This nomenclature could lead to confusion with address nomenclature.

On page 4, line 111, reference is made to addresses 11 and 22 (?)

It would be interesting, at the beginning or at the end of the Material and method, to add a table with the stress and direction settings used.

There are many references that do not appear correctly in the text.

The degrees in temperature are not well indicated (do not use º, it is not the correct symbol)
